# Learning Multiple Tasks in Parallel with a Shared Annotator

**Haim Cohen**
Department of Electrical Engeneering
The Technion – Israel Institute of Technology
Haifa, 32000 Israel
hcohen@tx.technion.ac.il

**Koby Crammer**
Department of Electrical Engeneering
The Technion – Israel Institute of Technology
Haifa, 32000 Israel
koby@ee.technion.ac.il

## Abstract

We introduce a new multi-task framework, in which $K$ online learners are sharing a single annotator with limited bandwidth. On each round, each of the $K$ learners receives an input, and makes a prediction about the label of that input. Then, a shared (stochastic) mechanism decides which of the $K$ inputs will be annotated. The learner that receives the feedback (label) may update its prediction rule, and then we proceed to the next round. We develop an online algorithm for multi-task binary classification that learns in this setting, and bound its performance in the worst-case setting. Additionally, we show that our algorithm can be used to solve two bandits problems: contextual bandits, and dueling bandits with context, both allow to decouple exploration and exploitation. Empirical study with OCR data, vowel prediction (VJ project) and document classification, shows that our algorithm outperforms other algorithms, one of which uses uniform allocation, and essentially achieves more (accuracy) for the same labour of the annotator.

## 1 Introduction

A triumph of machine learning is the ability to predict many human aspects: is certain mail spam or not, is a news-item of interest or not, does a movie meet one's taste or not, and so on. The dominant paradigm is supervised learning, in which the main bottleneck is the need to annotate data. A common protocol is problem centric: first collect data or inputs automatically (with low cost), and then pass it on to a user or an expert to be annotated. Annotation can be outsourced to the crowed by a service like Mechanical Turk, or performed by experts as in the Linguistic data Consortium. Then, this data may be used to build models, either for a single task or many tasks. This approach is not making optimal use of the main resource - the annotator - as some tasks are harder than others, yet we need to give the annotator the (amount of) data to be annotated for each task a-priori . Another aspect of this problem is the need to adapt systems to individual users, to this end, such systems may query the user for the label of some input, yet, if few systems will do so independently, the user will be flooded with queries, and will avoid interaction with those systems. For example, sometimes there is a need to annotate news items from few agencies. One person cannot handle all of them, and only some items can be annotated, which ones? Our setting is designed to handle exactly this problem, and specifically, how to make best usage of annotation time.

We propose a new framework of online multi-task learning with a shared annotator. Here, algorithms are learning few tasks simultaneously, yet they receive feedback using a central mechanism that trades off the amount of feedback (or labels) each task receives. We derive a specific algorithm based on the good-old Perceptron algorithm, called SHAMPO (SHared Annotator for Multiple PrOblems) for binary classification and analyze it in the mistake bound model, showing that our algorithm may perform well compared with methods that observe all annotated data. We then show how to reduce few contextual bandit problems into our framework, and provide specific bounds for such

settings. We evaluate our algorithm with four different datasets for OCR , vowel prediction (VJ) and document classification, and show that it can improve performance either on average over all tasks, or even if their output is combined towards a single shared task, such as multi-class prediction. We conclude with discussion of related work, and few of the many routes to extend this work.

## 2   Problem Setting

We study online multi-task learning with a shared annotator. There are $K$ tasks to be learned simultaneously. Learning is performed in rounds. On round $t$, there are $K$ input-output pairs $(\mathbf{x}_{i,t}, y_{i,t})$ where inputs $\mathbf{x}_{i,t} \in \mathbb{R}^{d_i}$ are vectors, and labels are binary $y_{i,t} \in \{-1, +1\}$. In the general case, the input-spaces for each task may be different. We simplify the notation and assume that $d_i = d$ for all tasks. Since the proposed algorithm uses the margin that is affected by the vector norm, there is a need to scale all the vectors into a ball. Furthermore, no dependency between tasks is assumed.

On round $t$, the learning algorithm receives $K$ inputs $\mathbf{x}_{i,t}$ for $i = 1, \ldots, K$, and outputs $K$ binary-labels $\hat{y}_{i,t}$, where $\hat{y}_{i,t} \in \{-1, +1\}$ is the label predicted for the input $\mathbf{x}_{i,t}$ of task $i$. The algorithm then chooses a task $J_t \in \{1, \ldots, K\}$ and receives from an annotator the true-label $y_{J_t,t}$ for that task $J_t$. It does not observe any other label.

Then, the algorithm updates its models, and proceeds to the next round (and inputs). For easing calculations below, we denote by $K$ indicators $Z_t = (Z_{1,t}, \ldots, Z_{K,t})$ the identity of the task which was queried on round $t$, and set $Z_{J_t,t} = 1$ and $Z_{i,t} = 0$ for $i \neq J_t$. Clearly, $\sum_i Z_{i,t} = 1$. Below, we define the notation $\mathrm{E}_{t-1}[x]$ to be the conditional expectation $\mathrm{E}[x|Z_1, \ldots Z_{t-1}]$ given all previous choices.

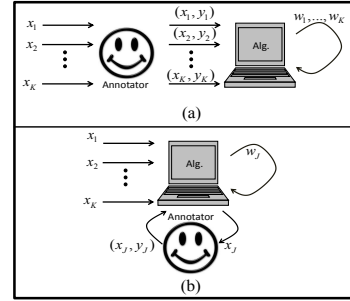
(a)

(b)

Figure 1: Illustration of a single iteration of multi-task algorithms (a) standard setting (b) SHAMPO

Illustration of a single iteration of multi-task algorithms is shown in Fig. 1. The top panel shows the standard setting with shared annotator, that labels all inputs, which are fed to the corresponding algorithms to update corresponding models. The bottom panel shows the SHAMPO algorithm, which couples labeling annotation and learning process, and synchronizes a single annotation per round. At most one task performs an update per round (the annotated one).

We focus on linear-functions of the form $f(\mathbf{x}) = \mathrm{sign}(p)$ for a quantity $p = \mathbf{w}^\top \mathbf{x}$, $\mathbf{w} \in \mathbb{R}^d$, called the margin. Specifically, the algorithm maintains a set of $K$ weight vectors. On round $t$, the algorithm predicts $\hat{y}_{i,t} = \mathrm{sign}(\hat{p}_{i,t})$ where $\hat{p}_{i,t} = \mathbf{w}_{i,t-1}^\top \mathbf{x}_{i,t}$. On rounds for which the label of some task $J_t$ is queried, the algorithm, is *not* updating the models of all *other* tasks, that is, we have $\mathbf{w}_{i,t} = \mathbf{w}_{i,t-1}$ for $i \neq J_t$.

We say that the algorithm has a prediction mistake in task $i$ if $y_{i,t} \neq \hat{y}_{i,t}$, and denote this event by $M_{i,t} = 1$, otherwise, if there is no mistake we set $M_{i,t} = 0$. The goal of the algorithm is to minimize the cumulative number of mistakes, $\sum_t \sum_i M_{i,t}$. Models are also evaluated using the Hinge-loss. Specifically, let $\mathbf{u}_i \in \mathbb{R}^d$ be some vector associated with task $i$. We denote the Hinge-loss of it, with respect to some input-output by, $\ell_{\gamma,i,t}(\mathbf{u}_i) = \left(\gamma - y_{i,t}\mathbf{u}_i^\top \mathbf{x}_{i,t}\right)_+$, where, $(x)_+ = \max\{x, 0\}$, and $\gamma > 0$ is some parameter. The cumulative loss over all tasks and a sequence of $n$ inputs, is, $L_{\gamma,n} = L_\gamma(\{\mathbf{u}_i\}) = \sum_{t=1}^n \sum_{i=1}^K \ell_{\gamma,i,t}(\mathbf{u}_i)$. We also use the following expected hinge-loss over the random choices of the algorithm, $\bar{L}_{\gamma,n} = \bar{L}_{\{\mathbf{u}_i\}} = \mathrm{E}\left[\sum_t^n \sum_{i=1}^K M_{i,t}Z_{i,t}\ell_{\gamma,i,t}(\mathbf{u}_i)\right]$. We proceed by describing our algorithm and specifying how to choose a task to query its label, and how to perform an update.

## 3   SHAMPO: SHared Annotator for Multiple Problems

We turn to describe an algorithm for multi-task learning with a shared annotator setting, that works with linear models. Two steps are yet to be specified: how to pick a task to be labeled and how to perform an update once the true label for that task is given.

To select a task, the algorithm uses the absolute margin $|\hat{p}_{i,t}|$. Intuitively, if $|\hat{p}_{i,t}|$ is small, then there is uncertainty about the labeling of $\mathbf{x}_{i,t}$, and vise-versa for large values of $|\hat{p}_{i,t}|$. Similar argument

was used by Tong and Koller [22] for picking an example to be labeled in batch active learning. Yet, if the model $\mathbf{w}_{i,t-1}$ is not accurate enough, due to small number of observed examples, this estimation may be rough, and may lead to a wrong conclusion. We thus perform an exploration-exploitation strategy, and query tasks randomly, with a bias towards tasks with low $|\hat{p}_{i,t}|$. To the best of our knowledge, exploration-exploitation usage in this context of choosing an examples to be labeled (eg. in settings such as semi-supervised learning or selective sampling) is novel and new. We introduce $b \geq 0$ to be a tradeoff parameter between exploration and exploitation and $a_i \geq 0$ as a prior for query distribution over tasks. Specifically, we induce a distribution over tasks,

$$\Pr\left[J_t = j\right] = \frac{a_j\left(b + |\hat{p}_{j,t}| - \min_{m=1}^{K}|\hat{p}_{m,t}|\right)^{-1}}{D_t} \quad \text{for } D_t = \sum_{i=1}^{K} a_i\left(b + |\hat{p}_{i,t}| - \min_m|\hat{p}_{m,t}|\right)^{-1}. \quad (1)$$

**Parameters:** $b, \lambda, a_i \in \mathbb{R}_+$ for $i = 1, \ldots, K$
**Initialize:** $\mathbf{w}_{i,0} = \mathbf{0}$ for $i = 1, \ldots, K$
**for** $t = 1, 2, ..., n$ **do**
  1. Observe $K$ instance vectors, $\mathbf{x}_{i,t}$, $(i = 1, \ldots, K)$.
  2. Compute margins $\hat{p}_{i,t} = \mathbf{w}_{i,t-1}^{\top}\mathbf{x}_{i,t}$.
  3. Predict $K$ labels, $\hat{y}_{i,t} = \text{sign}(\hat{p}_{i,t})$.
  4. Draw task $J_t$ with the distribution:

$$\Pr\left[J_t = j\right] = \frac{a_j\left(b + |\hat{p}_{j,t}| - \min_{m=1}^{K}|\hat{p}_{m,t}|\right)^{-1}}{D_t},$$

$$D_t = \sum_i a_i\left(b + |\hat{p}_{i,t}| - \min_{m=1}^{K}|\hat{p}_{m,t}|\right)^{-1}.$$

  5. Query the true label ,$y_{J_t,t} \in \{-1, 1\}$.
  6. Set indicator $M_{J_t,t} = 1$ iff $y_{J_t,t}\hat{p}_{i,t} \leq 0$ (Error)
  7. Set indicator $A_{J_t,t} = 1$ iff $0 < y_{J_t,t}\hat{p}_{i,t} \leq \lambda$ (Small margin)
  8. Update with the perceptron rule:

$$\mathbf{w}_{J_t,t} = \mathbf{w}_{J_t,t-1} + \left(A_{J_t,t} + M_{J_t,t}\right)y_{J_t,t}\mathbf{x}_{J_t,t} \quad (2)$$
$$\mathbf{w}_{i,t} = \mathbf{w}_{i,t-1} \text{ for } i \neq J_t$$

**end for**
**Output**: $\mathbf{w}_{i,n}$ for $i = 1, \ldots, K$.

Figure 2: SHAMPO: SHared Annotator for Multiple PrOblems.

Clearly, $\Pr\left[J_t = j\right] \geq 0$ and $\sum_j \Pr\left[J_t = j\right] = 1$. For $b = 0$ we have $\Pr\left[J_t = j\right] = 1$ for the task with minimal margin, $J_t = \arg\min_{i=1}^{K}|\hat{p}_{i,t}|$, and for $b \to \infty$ the distribution is proportional to the prior weights, $\Pr\left[J_t = j\right] = a_j/(\sum_i a_i)$. As noted above we denote by $Z_{i,t} = 1$ iff $i = J_t$. Since the distribution is invariant to a multiplicative factor of $a_i$ we assume $1 \leq a_i \forall i$.

The update of the algorithm is performed with the aggressive perceptron rule, that is $\mathbf{w}_{J_t,t} = \mathbf{w}_{J_t,t-1} + (A_{J_t,t} + M_{J_t,t})y_{J_t,t}\mathbf{x}_{J_t,t}$ and $\mathbf{w}_{i,t} = \mathbf{w}_{i,t-1}$ for $i \neq J_t$. we define $A_{i,t}$, the aggressive update indicator introducing and the aggressive update threshold, $\lambda \in \mathbb{R} > 0$ such that, $A_i = 1$ iff $0 < y_{i,t}\hat{p}_{i,t} \leq \lambda$, i.e, there is no mistake but the margin is small, and $A_{i,t} = 0$ otherwise. An update is performed if either there is a mistake ($M_{J_i,t} = 0$) or the margin is low ($A_{J_i,t} = 1$). Note that these events are mutually exclusive. For simplicity of presentation we write this update as, $\mathbf{w}_{i,t} = \mathbf{w}_{i,t-1} + Z_{i,t}(A_{i,t} + M_{i,t})y_{i,t}\mathbf{x}_{i,t}$. Although this notation uses labels for all-tasks, only the label of the task $J_t$ is used in practice, as for other tasks $Z_{i,t} = 0$.

We call this algorithm *SHAMPO* for SHared Annotator for Multiple PrOblems. The pseudo-code appears in Fig. 2. We conclude this section by noting that the algorithm can be incorporated with Mercer-kernels as all operations depend implicitly on inner-product between inputs.

## 4 Analysis

The following theorem states that the expected cumulative number of mistakes that the algorithm makes, may not be higher than the algorithm that observes the labels of all inputs.

**Theorem 1** *If SHAMPO algorithm runs on $K$ tasks with $K$ parallel example pair sequences $(\mathbf{x}_{i,1}, y_{i,1}), ...(\mathbf{x}_{i,n}, y_{i,n}) \in \mathbb{R}^d \times \{-1, 1\}$, $i = 1, ..., K$ with input parameters $0 \leq b$, $0 \leq \lambda \leq b/2$, and prior $1 \leq a_i \forall i$, denote by $X = \max_{i,t}\|\mathbf{x}_{i,t}\|$, then, for all $\gamma > 0$, all $\mathbf{u}_i \in \mathbb{R}^d$ and all $n \geq 1$*

*there exists $0 < \delta \le \sum_{i=1}^{K} a_i$, such that,*

$$\mathbb{E}\left[\sum_{i=1}^{K}\sum_{t=1}^{n} M_{i,t}\right] \le \frac{\delta}{\gamma}\left[\left(1 + \frac{X^2}{2b}\right)\bar{L}_{\gamma,n} + \frac{(2b + X^2)^2 U^2}{8\gamma b}\right] + \left(2\frac{\lambda}{b} - 1\right)\mathbb{E}\left[\sum_{i=1}^{K}\sum_{t=1}^{n} a_i A_{i,t}\right].$$

*where we denote $U^2 = \sum_{i=1}^{K} \|\mathbf{u}_i\|^2$. The expectation is over the random choices of the algorithm.*

Due to lack of space, the proof appears in Appendix A.1 in the supplementary material. Few notes on the mistake bound: First, the right term of the bound is equals zero either when $\lambda = 0$ (as $A_{i,t} = 0$) or $\lambda = b/2$. Any value in between, may yield an strict negative value of this term, which in turn, results in a lower bound. Second, the quantity $\bar{L}_{\gamma,n}$ is *non*-increasing with the number of tasks. The first terms depends on the number of tasks only via $\delta \le \sum_i a_i$. Thus, if $a_i = 1$ (uniform prior) the quantity $\delta \le K$ is bounded by the number of tasks. Yet, when the hardness of the tasks is not equal or balanced, one may expect $\delta$ to be closer to 1 than $K$, which we found empirically to be true. Additionally, the prior $a_i$ can be used to make the algorithm focus on the hard tasks, thereby improving the bound. While $\delta$ multiplying the first term can be larger, the second term can be lower. A task $i$ which corresponds to a large value of $a_i$ will be updated more in early rounds than tasks with low $a_i$. If more of these updates are aggressive, the second term will be negative and far from zero.

One can use the bound to tune the algorithm for a good value of $b$ for the non aggressive case ($\lambda = 0$), by minimizing the bound over $b$. This may not be possible directly since $\bar{L}_{\gamma,n}$ depends implicitly on the value of $b$[1]. Alternatively, we can take a loose estimate of $\bar{L}_{\gamma,n}$, and re-place it with $L_{\gamma,n}$ (which is $\sim K$ times larger). The optimal value of $b$ can now be calculated, $b = \frac{X^2}{2}\sqrt{1 + \frac{4\gamma L_{\gamma,n}}{U^2 X^2}}$. Substituting this value in the bound of Eq. (1) with $L_{\gamma,n}$ leads to the following bound, $\mathbb{E}\left[\sum_{i=1}^{K}\sum_{t=1}^{n} M_{i,t}\right] \le \frac{\delta}{\gamma}\left[L_{\gamma,n} + \frac{U^2 X^2}{2\gamma} + \frac{U^2}{2\gamma}\sqrt{1 + \frac{4\gamma L_{\gamma,n}}{U^2 X^2}}\right]$, which has the same dependency in the number of inputs $n$ as algorithm that observes all of them.

We conclude this section by noting that the algorithm and analysis can be extended to the case that more than single query is allowed per task. Analysis and proof appears in Appendix A.2 in the supplementary material.

## 5  From Multi-task to Contextual Bandits

Although our algorithm is designed for many binary-classification tasks, it can also be applied in two settings of contextual bandits, when decoupling exploration and exploitation is allowed [23, 3]. In this setting, the goal is to predict a label $\hat{Y}_t \in \{1, \dots, C\}$ given an input $\mathbf{x}_t$. As before, the algorithm works in rounds. On round $t$ the algorithm receives an input $\mathbf{x}_t$ and gives as an output multicalss label $\hat{Y}_t \in \{1, \dots, C\}$. Then, it queries for some information about the label via a single binary "yes-no" question, and uses the feedback to update its model. We consider two forms of questions. Note that our algorithm subsumes past methods since they also allow the introduction of a bias (or prior knowledge) towards some tasks, which in turn, may improve performance.

### 5.1  One-vs-Rest
The first setting is termed *one-vs-rest*. The algorithm asks if the true label is some label $\bar{Y}_t \in \{1, \dots, C\}$, possibly not the predicted label, i.e. it may be the case that $\bar{Y}_t \ne \hat{Y}_t$. Given the response whether $\bar{Y}_t$ is the true label $Y_t$, the algorithm updates its models. The reduction we perform is by introducing $K$ tasks, one per class. The problem of the learning algorithm for task $i$ is to decide whether the true label is class $i$ or not. Given the output of all (binary) classifiers, the algorithm generates a single multi-class prediction to be the single label for which the output of the corresponding binary classifier is positive. If such class does not exist, or there are more than one classes as such, a random prediction is used, i.e., given an input $\mathbf{x}_t$ we define $\hat{Y}_t = \arg\max_i \hat{y}_{i,t}$, where ties are broken arbitrarily. The label to be queried is $\bar{Y}_t = J_t$, i.e. the problem index that SHAMPO is querying. We analyze the performance of this reduction as a multiclass prediction algorithm.

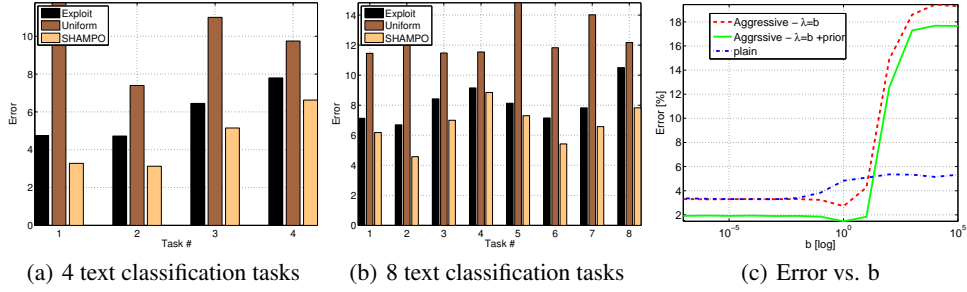

(a) 4 text classification tasks     (b) 8 text classification tasks     (c) Error vs. b

Figure 3: Left and middle: Test error of aggressive SHAMPO on (a) four and (b) eight binary text classification tasks. Three algorithms are evaluated: uniform, exploit, and aggressive SHAMPO. (Right) Mean test error over USPS One-vs-One binary problems vs $b$ of aggressive SHAMPO with prior, aggressive with uniform prior, and non-aggressive with uniform prior.

**Corollary 2** *Assume the SHAMPO algorithm is executed as above with $K = C$ one-vs-rest problems, on a sequence $(\mathbf{x}_1, Y_1), ...(\mathbf{x}_n, Y_n) \in \mathbb{R}^d \times \{1, ..., C\}$, and input parameter $b > 0$ and prior $1 \leq a_i \forall i$. Then for all $\gamma > 0$ and all $\mathbf{u}_i \in \mathbb{R}^d$, there exist $0 < \delta \leq \sum_{i=1}^C a_i$ such that the expected number of multi-class errors is bounded as follows $\mathbb{E}\left[\sum_t \llbracket Y_t \neq \hat{Y}_t \rrbracket \right] \leq$*

$$\frac{\delta}{\gamma}\left[\left(1 + \frac{X^2}{2b}\right)\bar{L}_{\gamma,n} + \frac{(2b+X^2)^2 U^2}{8\gamma b}\right] + \left(2\frac{\lambda}{b} - 1\right)\mathbb{E}\left[\sum_{i=1}^K \sum_{t=1}^n a_i A_{i,t}\right] \text{, where } \llbracket I \rrbracket = 1 \text{ if the pred-}$$

*icate I is true, and zero otherwise.*

The corollary follows directly from Thm. 1 by noting that, $\llbracket Y_t \neq \hat{Y}_t \rrbracket \leq \sum_i M_{i,t}$. That is, there is a multiclass mistake if there is at least one prediction mistake of one of the one-vs-rest problems. The closest setting is contextual bandits, yet we allow decoupling of exploration and exploitation. Ignoring this decoupling, the Banditron algorithm [17] is the closest to ours, with a regret of $O(T^{2/3})$. Hazan et al [16] proposed an algorithm with $O(\sqrt{T})$ regret but designed for the $\log$ loss, with coefficient that may be very large, and another [9] algorithm has $O(\sqrt{T})$ regret with respect to prediction mistakes, yet they assumed stochastic labeling, rather than adversarial.

### 5.2 One-vs-One

In the second setting, termed by *one-vs-one*, the algorithm picks two labels $\bar{Y}_t^+, \bar{Y}_t^- \in \{1 ... C\}$, possibly both not the predicted label. The feedback for the learner is three-fold: it is $y_{J_t,t} = +1$ if the first alternative is the correct label, $\bar{Y}_t^+ = Y_t$, $y_{J_t,t} = -1$ if the second alternative is the correct label, $\bar{Y}_t^- = Y_t$, and it is $y_{J_t,t} = 0$ otherwise (in this case there is no error and we set $M_{J_t,t} = 0$). The reduction we perform is by introducing $K = \binom{C}{2}$ problems, one per pair of classes. The goal of the learning algorithm for a problem indexed with two labels $(y_1, y_2)$ is to decide which is the correct label, given it is one of the two. Given the output of all (binary) classifiers the algorithm generates a single multi-class prediction using a tournament in a round-robin approach [15]. If there is no clear winner, a random prediction is used. We now analyze the performance of this reduction as a multiclass prediction algorithm.

**Corollary 3** *Assume the SHAMPO algorithm is executed as above, with $K = \binom{C}{2}$ one-vs-one problems, on a sequence $(\mathbf{x}_1, Y_1), ...(\mathbf{x}_n, Y_n) \in \mathbb{R}^d \times \{1, ..., C\}$, and input parameter $b > 0$ and prior $1 \leq a_i \forall i$. Then for all $\gamma > 0$ and all $\mathbf{u}_i \in \mathbb{R}^d$, there exist $0 < \delta \leq \sum_{i=1}^{\binom{C}{2}} a_i$ such that the expected number of multi-class errors can be bounded as follows $\mathbb{E}\left[\sum_t \llbracket Y_t \neq \hat{Y}_t \rrbracket \right] \leq$*

$$\frac{2}{(\binom{C}{2}-1)/2+1}\left\{\frac{\delta}{\gamma}\left[\left(1 + \frac{X^2}{2b}\right)\bar{L}_{\gamma,n} + \frac{(2b+X^2)^2 U^2}{8\gamma b}\right] + \left(2\frac{\lambda}{b} - 1\right)\mathbb{E}\left[\sum_{i=1}^K \sum_{t=1}^n a_i A_{i,t}\right]\right\}.$$

The corollary follows directly from Thm. 1 by noting that, $\llbracket Y_t \neq \hat{Y}_t \rrbracket \leq \frac{2}{(\binom{C}{2}-1)/2+1}\sum_{i=1}^{\binom{C}{2}} M_{i,t}$. Note, that the bound is essentially independent of $C$ as the coefficient in the bound is upper bounded by 6 for $C \geq 3$.

We conclude this section with two algorithmic modifications, we employed in this setting. Currently, when the feedback is zero, there is no update of the weights, because there are no errors. This causes the algorithm to effectively ignore such examples, as in these cases the algorithm is not modifying any model, furthermore, if such example is repeated, a problem with possibly "0" feedback may be queried again. We fix this issue with one of two modifications: In the first one, if the feedback is zero, we modify the model to reduce the chance that the chosen problem, $J_t$, would be chosen again for the same input (i.e. not to make the same wrong-choice of choosing irrelevant problem again). To this end, we modify the weights a bit, to increase the confidence (absolute margin) of the model for the same input, and replace Eq. (2) with, $\mathbf{w}_{J_t,t} = \mathbf{w}_{J_t,t-1} + [\![y_{J_t,t} \neq 0]\!]\, y_{J_t,t}\, \mathbf{x}_{J_t,t} + [\![y_{J_t,t} = 0]\!]\eta \hat{y}_{J_t,t}\mathbf{x}_{J_t,t}$ , for some $\eta > 0$. In other words, if there is a possible error (i.e. $y_{J_t,t} \neq 0$) the update follows the Perceptron's rule. Otherwise, the weights are updated such that the absolute margin will increase, as $|\mathbf{w}_{J_t,t}^\top \mathbf{x}_{J_t,t}| = |(\mathbf{w}_{J_t,t-1} + \eta \hat{y}_{J_t,t}\mathbf{x}_{J_t,t})^\top \mathbf{x}_{J_t,t}| = |\mathbf{w}_{J_t,t-1}^\top \mathbf{x}_{J_t,t} + \eta \mathrm{sign}(\mathbf{w}_{J_t,t-1}^\top \mathbf{x}_{J_t,t})\|\mathbf{x}_{J_t,t}\|^2| = |\mathbf{w}_{J_t,t-1}^\top \mathbf{x}_{J_t,t}| + \eta\|\mathbf{x}_{J_t,t}\|^2 > |\mathbf{w}_{J_t,t-1}^\top \mathbf{x}_{J_t,t}|$. We call this method *one-vs-one-weak*, as it performs weak updates for zero feedback. The second alternative is not to allow 0 value feedback, and if this is the case, to set the label to be either $+1$ or $-1$, randomly. We call this method *one-vs-one-random*.

## 6 Experiments

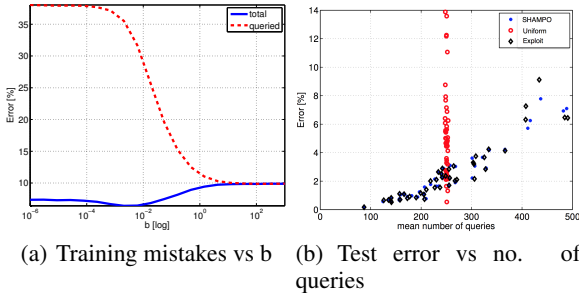

(a) Training mistakes vs b (b) Test error vs no. of queries

Figure 4: Left: mean of fraction no. of mistakes SHAMPO made during training time on MNIST of all examples and only queried. Right: test error vs no. of queries is plotted for all MNIST one-vs-one problems.

We evaluated the SHAMPO algorithm using four datasets: USPS, MNIST (both OCR), Vocal Joystick (VJ, vowel recognition) and document classification. The USPS dataset, contains $7,291$ training examples and $2,007$ test examples, each is a $16 \times 16$ pixels gray-scale images converted to a 256 dimensional vector. The MNIST dataset with $28 \times 28$ gray-scale images, contains $60,000$ $(10,000)$ training (test) examples. In both cases there are 10 possible labels, digits. The VJ tasks is to predict a vowel from eight possible vowels. Each example is a frame of spoken value described with 13 MFCC coefficients transformed into 27 features. There are $572,911$ training examples and $236,680$ test examples. We created binary tasks from these multi-class datasets using two reductions: One-vs-Rest setting and One-vs-One setting. For example, in both USPS and MNIST there are 10 binary one-vs-rest tasks and 45 binary one-vs-one tasks. The NLP document classification include of spam filtering, news items and news-group classification, sentiment classification, and product domain categorization. A total of 31 binary prediction tasks over all, with a total of $252,609$ examples, and input dimension varying between $8,768$ and $1,447,866$. Details of the individual binary tasks can be found elsewhere [8]. We created an eighth collection, named MIXED, which consists of 40 tasks: 10 random tasks from each one of the four basic datasets (one-vs-one versions). This yielded eight collections (USPS, MNIST and VJ; each as one-vs-rest or one-vs-one), document classification and mixed. From each of these eight collections we generated between 6 to 10 combinations (or problems), each problem was created by sampling between 2 and 8 tasks which yielded a total of 64 multi-task problems. We tried to diversify problems difficulty by including both hard and easy binary classification problems. The hardness of a binary problem is evaluated by the number of mistakes the Perceptron algorithm performs on the problem.

We evaluated two baselines as well as our algorithm. Algorithm *uniform* picks a random task to be queried and updated (corresponding to $b \to \infty$), *exploit* which picks the tasks with the lowest absolute margin (i.e. the "hardest instance"), this combination corresponds to $b \approx 0$ of SHAMPO. We tried for SHAMPO 13 values for $b$, equally spaced on a logarithmic scale. All algorithms made a single pass over the training data. We ran two version of the algorithm: plain version, without aggressiveness (updates on mistakes only, $\lambda = 0$) and an Aggressive version $\lambda = b/2$ (we tried lower values of $\lambda$ as in the bound, but we found that $\lambda = b/2$ gives best results), both with uniform prior ($a_i = 1$). We used separate training set and a test set, to build a model and evaluate it.

Table 1: Test errors percentage . Scores are shown in parenthesis.

| Dataset | Aggressive $\lambda = b/2$ | | | Plain | | |
|---|---|---|---|---|---|---|
| | *exploit* | *SHAMPO* | *uniform* | *exploit* | *SHAMPO* | *uniform* |
| VJ 1 vs 1 | 5.22 (2.9) | **4.57 (1.1)** | 5.67 (3.9) | 5.21 (2.7) | 6.93 (4.6) | 6.26 (5.8) |
| VJ 1 vs Rest | 13.26 (3.5) | **11.73 (1.2)** | 12.43 (2.5) | 13.11 (3.0) | 14.17 (5.0) | 14.71 (5.8) |
| USPS 1 vs 1 | 3.31 (2.5) | **2.73 (1.0)** | 19.29 (6.0) | 3.37 (2.5) | 4.83 (4.0) | 5.33 (5,0) |
| USPS 1 vs Rest | 5.45 (2.8) | **4.93 (1.2)** | 10.12 (6.0) | 5.31 (2.0) | 6.51 (4.0) | 7.06 (5.0) |
| MNIST 1 vs 1 | 1.08 (2.3) | **0.75 (1.0)** | 5.9 (6.0) | 1.2 (2.7) | 1.69 (4.1) | 1.94 (4.9) |
| MNIST 1 vs Rest | 4.74 (2.8) | **3.88 (1.0)** | 10.01 (6.0) | 4.44 (2.8) | 5.4 (3.8) | 6.1 (5.0) |
| NLP documents | 19.43 (2.3) | **16.5 (1.0)** | 23.21 (5.0) | 19.46 (2.7) | 21.54 (4.7) | 21.74 (5.3) |
| MIXED | 2.75 (2.4) | **2.06 (1.0)** | 13.59 (6.0) | 2.78 (2.6) | 4.2 (4.3) | 4.45 (4.7) |
| *Mean score* | (2.7) | **(1.1)** | (5.2) | (2.6) | (4.3) | (5.2) |

Results are evaluated using 2 quantities. First, the average test error (over all the dataset combinations) and the average score. For each combination we assigned a score of 1 to the algorithm with the lowest test error, and a score of 2, to the second best, and all the way up to a score of 6 to the algorithm with the highest test error.

**Multi-task Binary Classification :**   Fig. 3(a) and Fig. 3(b) show the test error of the three algorithms on two of document classification combinations, with four and eight tasks. Clearly, not only SHAMPO performs better, but it does so on each task individually. (Our analysis above bounds the total number of mistakes over all tasks.) Fig. 3(c) shows the average test error vs $b$ using the one-vs-one binary USPS problems for the three variants of SHAMPO: non-aggressive (called plain), aggressive and aggressive with prior.Clearly, the plain version does worse than both the aggressive version and the non-uniform prior version. For other combinations the prior was not always improving results. We hypothesise that this is because our heuristic may yield a bad prior which is not focusing the algorithm on the right (hard) tasks.

Results are summarized in Table 1. In general *exploit* is better than *uniform* and aggressive is better than non-aggressive. Aggressive SHAMPO yields the best results both evaluated as average (over tasks per combination and over combinations). Remarkably, even in the mixed dataset (where tasks are of different nature: images, audio and documents), the aggressive SHAPO improves over uniform (4.45% error) and the aggressive-exploit baseline (2.75%), and achieves a test error of 2.06%.

Next, we focus on the problems that the algorithm chooses to annotate on each iteration for various values of $b$. Fig. 4(a) shows the total number of mistakes SHAMPO made during training time on MNIST , we show two quantities: fraction of mistakes over all training examples (denoted by "total" - blue) and fraction of mistakes over only queried examples (denoted by "queried" - dashed red). In pure exploration (large $b$) both quantities are the same, as the choice of problem to be labeled is independent of the problem and example, and essentially the fraction of mistakes in queried examples is a good estimate of the fraction of mistakes over all examples. The other extreme is when performing pure exploitation (low $b$), here the fraction of mistakes made on queried examples went up, while the overall fraction of mistakes went down. This indicates that the algorithm indeed focuses its queries on the harder inputs, which in turn, improves overall training mistake. There is a sweet point $b \approx 0.01$ for which SHAMPO is still focusing on the harder examples, yet reduces the total fraction of training mistakes even more. The existence of such tradeoff is predicted by Thm. 1.

Another perspective of the phenomena is that for values of $b \ll \infty$ SHAMPO focuses on the harder examples, is illustrated in Fig. 4(b) where test error vs number of queries is plotted for each problem for MNIST. We show three cases: uniform, exploit and a mid-value of $b \approx 0.01$ which tradeoffs exploration and exploitation. Few comments: First, when performing uniform querying, all problems have about the same number of queries (266), close to the number of examples per problem (12,000), divided by the number of problems (45). Second, when having a tradeoff between exploration and exploitation, harder problems (as indicated by test error) get more queries than easier problems. For example, the four problems with test error greater than 6% get at least 400 queries, which is about twice the number of queries received by each of the 12 problems with test error less than 1%. Third, as a consequence, SHAMPO performs equalization, giving the harder problems more labeled data, and as a consequence, reduces the error of these problems, however, is not increasing the error of the easier problems which gets less queries (in fact it reduces the test error of all 45 problems!). The tradeoff mechanism of SHAMPO, reduces the test error of each problem

by more than $40\%$ compared to full exploration. Fourth, exploits performs similar equalization, yet in some hard tasks it performs worse than SHAMPO. This could be because it overfits the training data, by focusing on hard-examples too much, as SHAMPO has a randomness mechanism.

Indeed, Table 1 shows that aggressive SHAMPO outperforms better alternatives. Yet, we claim that a good prior may improve results. We compute prior over the 45 USPS tasks, by running the perceptron algorithm on 1000 examples and computing the number of mistakes. We set the prior to be proportional to this number. We then reran aggressive SHAMPO with prior, comparing it to aggressive SHAMPO with no prior (i.e. $a_i = 1$). Aggressive SHAMO with prior achieves average error of $1.47$ (vs. $2.73$ with no prior) on 1-vs-1 USPS and $4.97$ (vs $4.93$) on one-vs-rest USPS, with score rank of $1.0$ (vs $2.9$) and $1.7$ (vs $2.0$) respectively. Fig. 3(c) shows the test error for a all values of $b$ we evaluated. A good prior is shown to outperform the case $a_i = 1$ for all values of $b$.

**Reduction of Multi-task to Contextual Bandits**     Next, we evaluated SHAMPO as a contextual bandit algorithm, by breaking a multi-class problem into few binary tasks, and integrating their output into a single multi-class problem. We focus on the VJ data, as there are many examples, and linear models perform relatively well [18]. We implemented all three reductions mentioned in Sec. 5.2, namely, *one-vs-rest*, *one-vs-one-random* which picks a random label if the feedback is zero, *one-vs-one-weak* (which performs updates to increase confidence when the feedback is zero), where we set $\eta = 0.2$, and the Banditron algorithm [17]. The *one-vs-rest* reduction and the Banditron have a test error of about $43.5\%$, and the *one-vs-one-random* of about $42.5\%$. Finally, *one-vs-one-weak* achieves an error of $39.4\%$. This is slightly worst than PLM  [18] with test error of $38.4\%$ (and higher than MLP with $32.8\%$), yet all of these algorithms observe only one bit of feedback per example, while both MLP and PLM observe 3 bits (as class identity can be coded with 3 bits for 8 classes). We claim that our setting can be easily used to adapt a system to individual user, as we only need to assume the ability to recognise three words, such as three letters. Given an utterance of the user, the system may ask: "Did you say (a) 'a' like 'bad' (b) 'o' like in 'book') (c) none". The user can communicate the correct answer with no need for a another person to key in the answer.

# 7   Related Work and Conclusion

In the past few years there is a large volume of work on multi-task learning, which clearly we can not cover here. The reader is referred to a recent survey on the topic [20]. Most of this work is focused on exploring relations between tasks, that is, find similarities dissimilarities between tasks, and use it to share data directly (e.g. [10]) or model parameters [14, 11, 2]. In the online settings there are only a handful of work on multi-task learning. Dekel et al [13] consider the setting where all algorithms are evaluated using a global loss function, and all work towards the shared goal of minimizing it. Logosi et al [19] assume that there are constraints on the predictions of all learners, and focus in the expert setting. Agarwal et al [1] formalize the problem in the framework of stochastic convex programming with few matrix regularization, each captures some assumption about the relation between the models. Cavallanti et al  [4] and Cesa-Bianci et al [6] assume a known relation between tasks which is exploited during learning. Unlike these approaches, we assume the ability to share an annotator rather than data or parameters, thus our methods can be applied to problems that do not share a common input space.

Our analysis is similar to that of Cesa-Bianchi et al [7], yet they focus in selective sampling (see also [5, 12]), that is, making individual binary decisions of whether to query, while our algorithm always query, and needs to decide for which task. Finally, there have been recent work in contextual bandits [17, 16, 9], each with slightly different assumptions. To the best of our knowledge, we are the first to consider decoupled exploration and exploitation in this context. Finally, there is recent work in learning with relative or preference feedback in various settings [24, 25, 26, 21]. Unlike this work, our work allows again decoupled exploitation and exploration, and also non-relevant feedback.

To conclude, we proposed a new framework for online multi-task learning, where learners share a single annotator. We presented an algorithm (SHAMPO) that works in this settings and analyzed it in the mistake-bound model. We also showed how learning in such a model can be used to learn in contextual-bandits setting with few types of feedback. Empirical results show that our algorithm does better for the same price. It focuses the annotator on the harder instances, and is improving performance in various tasks and settings. We plan to integrate other algorithms to our framework, extend it to other settings, investigate ways to generate good priors, and reduce multi-class to binary also via error-correcting output-codes.

## Footnotes

[1]Similar issue appears also after the discussion of Theorem 1 in a different context [7].

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
