[Supplementary Material]

# A Supplementary Material

## A.1 Proof of Thm. 1

**Proof:** Fix $n$ examples sequences, $(\mathbf{x}_{i,1}, y_{i,1}), ..., (\mathbf{x}_{i,n}, y_{i,n})$ for each of the $K$ tasks. Let $t$ be certain trial and $i$ to be an update task on this trial, such that $M_{i,t} = 1$ or $A_{i,t} = 1$. Denote this event by $U_{i,t} = 1$. We write,

$$
\begin{aligned}
\gamma - \ell_{\gamma,i,t}(\mathbf{u}_i) &= \gamma - \left(\gamma - y_{i,t}\mathbf{u}_i^T\mathbf{x}_{i,t}\right)_+ \\
&\leq y_{i,t}\mathbf{u}_i^T\mathbf{x}_{i,t} \\
&= y_{i,t}\left(\mathbf{u}_i + \mathbf{w}_{i,t-1} - \mathbf{w}_{i,t-1}\right)^T\mathbf{x}_{i,t} \\
&= y_{i,t}\mathbf{w}_{i,t-1}^T\mathbf{x}_{i,t} + \frac{1}{2}\|\mathbf{u}_i - \mathbf{w}_{i,t-1}\|^2 \\
&\quad - \frac{1}{2}\|\mathbf{u}_i - \mathbf{w}_{i,t}\|^2 + \frac{1}{2}\|\mathbf{w}_{i,t-1} - \mathbf{w}_{i,t}\|^2 \\
&= y_{i,t}\hat{p}_{i,t} + \frac{1}{2}\|\mathbf{u}_i - \mathbf{w}_{i,t-1}\|^2 \\
&\quad - \frac{1}{2}\|\mathbf{u}_i - \mathbf{w}_{i,t}\|^2 + \frac{1}{2}\|\mathbf{w}_{i,t-1} - \mathbf{w}_{i,t}\|^2 .
\end{aligned}
$$

The last inequality holds for all $\gamma > 0$ and for all $\mathbf{u}_i \in \mathbb{R}^d$, so we can replace $\gamma$ and $\mathbf{u}_i$ by their scaling $\alpha\gamma$ and $\alpha\mathbf{u}_i$ respectively, where $\alpha > 0$ will be determined shortly and we get

$$\alpha\gamma + y_{i,t}\hat{p}_{i,t} \leq \alpha\ell_{\gamma,i,t}(\mathbf{u}_i) + \frac{1}{2}\|\alpha\mathbf{u}_i - \mathbf{w}_{i,t-1}\|^2 - \frac{1}{2}\|\alpha\mathbf{u}_i - \mathbf{w}_{i,t}\|^2 + \frac{1}{2}\|\mathbf{w}_{i,t-1} - \mathbf{w}_{i,t}\|^2.$$

In trials and task where there is no update, i.e., $U_{i,t}Z_{i,t} = 0$, the equality $\mathbf{w}_{i,t} = \mathbf{w}_{i,t-1}$ holds. Combining the last two observations, we have

$$U_{i,t}Z_{i,t}(\alpha\gamma + y_{i,t}\hat{p}_{i,t}) \leq U_{i,t}Z_{i,t}\alpha\ell_{\gamma,i,t}(\mathbf{u}_i) + \frac{1}{2}\|\alpha\mathbf{u}_i - \mathbf{w}_{i,t-1}\|^2 - \frac{1}{2}\|\alpha\mathbf{u}_i - \mathbf{w}_{i,t}\|^2 + \frac{1}{2}\|\mathbf{w}_{i,t-1} - \mathbf{w}_{i,t}\|^2.$$

Next, we sum the inequality above, over $t$ and use the fact that $\mathbf{w}_{i,0} = 0$ and $\|\mathbf{w}_{i,t-1} - \mathbf{w}_{i,t}\|^2 \leq X^2$ to get,

$$\sum_{t=1}^{n} U_{i,t}Z_{i,t}\left(\alpha\gamma + y_{i,t}\hat{p}_{i,t} - \frac{X^2}{2}\right) \leq \alpha\sum_{t=1}^{n} U_{i,t}Z_{i,t}\ell_{\gamma,i,t}(\mathbf{u}_i) + \frac{\alpha^2}{2}\|\mathbf{u}_i\|^2. \qquad (3)$$

Substituting $\alpha = (2b + X^2)/2\gamma$ (where $b \in \mathbb{R}, \; b > 0$) in Eq. (3), we get

$$\sum_{t=1}^{n} U_{i,t}Z_{i,t}(b + y_{i,t}\hat{p}_{i,t}) \leq \frac{2b + X^2}{2\gamma}\sum_{t=1}^{n} U_{i,t}Z_{i,t}\ell_{\gamma,i,t}(\mathbf{u}_i) + \frac{(2b + X^2)^2}{8\gamma^2}\|\mathbf{u}_i\|^2.$$

We subtract a non negative quantity $\sum_{t=1}^{n} U_{i,t}Z_{i,t}\min_j|\hat{p}_{j,t}|$ from the l.h.s. and get,

$$\sum_{t=1}^{n} U_{i,t}Z_{i,t}\left(b + y_{i,t}\hat{p}_{i,t} - \min_j|\hat{p}_{j,t}|\right) \leq \frac{2b + X^2}{2\gamma}\sum_{t=1}^{n} U_{i,t}Z_{i,t}\ell_{\gamma,i,t}(\mathbf{u}_i) + \frac{(2b + X^2)^2}{8\gamma^2}\|\mathbf{u}_i\|^2. \quad (4)$$

At this point we take the expectation of all the terms. Recall that the conditional expectation of $Z_{i,t}$ is $a_i(b + |\hat{p}_{i,t}| - \min_j|\hat{p}_{j,t}|)^{-1}/D_t$ and that $U_{i,t} = M_{i,t} + A_{i,t}$ and $\hat{p}_{i,t}$ are measurable with respect to the $\sigma$-algebra that generated by $Z_1, ...Z_{t-1}$. We start with the left term,

$$
\begin{aligned}
&\mathbb{E}\left[\sum_{t=1}^{n} U_{i,t}Z_{i,t}\left(b - y_{i,t}\hat{p}_{i,t} - \min_j|\hat{p}_{j,t}|\right)\right] \\
&= \mathbb{E}\left[\mathbb{E}_{t-1}\left[\sum_{t=1}^{n} U_{i,t}Z_{i,t}\left(b - y_{i,t}\hat{p}_{i,t} - \min_j|\hat{p}_{j,t}|\right)\right]\right] \\
&= \mathbb{E}\left[\sum_{t=1}^{n} \frac{a_i}{D_t}\left(M_{i,t} + \frac{b - |\hat{p}_{j,t}| - \min_j|\hat{p}_{j,t}|}{b + |\hat{p}_{j,t}| - \min_j|\hat{p}_{j,t}|}A_{i,t}\right)\right] .
\end{aligned}
$$

We remind the reader that $a_i \geq 1 \ \forall i$. Thus we bound $M_{i,t} \leq M_{i,t} a_i$ and get,

$$\mathbb{E}\left[\sum_{t=1}^{n} \frac{1}{D_t}\left(M_{i,t} + \frac{b - |\hat{p}_{j,t}| - \min_j |\hat{p}_{j,t}|}{b + |\hat{p}_{j,t}| - \min_j |\hat{p}_{j,t}|} A_{i,t} a_i\right)\right] \leq$$
$$\frac{2b + X^2}{2\gamma}\bar{L}_{\gamma,i,n}(u_i) + \frac{(2b + X^2)^2}{8\gamma^2}\|\mathbf{u}_i\|^2 . \tag{5}$$

Next we bound the factor that multiplies $a_i A_{i,t}$ as follows,

$$\left(1 - 2\frac{\lambda}{b}\right) = \frac{b - 2\lambda}{b} \leq \frac{b - |\hat{p}_{j,t}| - \min_j |\hat{p}_{j,t}|}{b + |\hat{p}_{j,t}| - \min_j |\hat{p}_{j,t}|},$$

and plug it into the left side of the inequality,

$$\mathbb{E}\left[\sum_{t=1}^{n} \frac{1}{D_t}\left(M_{i,t} + \frac{b - |\hat{p}_{j,t}| - \min_j |\hat{p}_{j,t}|}{b + |\hat{p}_{j,t}| - \min_j |\hat{p}_{j,t}|} A_{i,t} a_i\right)\right] \leq \mathbb{E}\left[\sum_{t=1}^{n} \frac{1}{D_t}\left(M_{i,t} + \left(1 - 2\frac{\lambda}{b}\right) A_{i,t} a_i\right)\right].$$

Since $b/2 \geq \lambda$ we have that $M_{i,t} + \left(1 - 2\frac{\lambda}{b}\right) A_{i,t} a_i \geq 0$, thus there exists $\delta_i$ such that,

$$\mathbb{E}\left[\sum_{t=1}^{n} \frac{1}{D_t}\left(M_{i,t} + \left(1 - 2\frac{\lambda}{b}\right) A_{i,t} a_i\right)\right] = \frac{b}{\delta_i}\mathbb{E}\left[\sum_{t=1}^{n}\left(M_{i,t} + \left(1 - 2\frac{\lambda}{b}\right) A_{i,t} a_i\right)\right], \tag{6}$$

and

$$\frac{b}{\delta_i} \geq \min \frac{1}{D_t} = \frac{1}{\max D_t} \geq \frac{b}{\sum_{j=1}^{K} a_j},$$

where the last inequality follows from,

$$D_t = \sum_{j=1}^{K} \frac{a_j}{\left(b + |\hat{p}_{j,t}| - \min_{m=1}^{K} |\hat{p}_{m,t}|\right)} \leq \frac{\sum_{j=1}^{K} a_j}{b}. \tag{7}$$

The previous bound implies that $0 < \delta_i \leq \sum_{i=1}^{K} a_i$.

Combining Eq. (5) and Eq. (6),

$$\frac{b}{\delta_i}\mathbb{E}\left[\sum_{t=1}^{n} M_{i,t}\right] \leq \frac{2b + X^2}{2\gamma}\bar{L}_{\gamma,i,n}(\mathbf{u}_i) + \frac{(2b + X^2)^2}{8\gamma^2}\|\mathbf{u}_i\|^2 + \frac{b}{\delta}\left(2\frac{\lambda}{b} - 1\right)a_i\mathbb{E}\left[\sum_{t=1}^{n} A_{i,t}\right]. \tag{8}$$

Summing up the last inequality over all $K$ tasks and setting $\delta = \max_i \delta_i$ yields,

$$\mathbb{E}\left[\sum_{i=1}^{K}\sum_{t=1}^{n} M_{i,t}\right] \leq \frac{\delta}{\gamma}\left[\left(1 + \frac{X^2}{2b}\right)\bar{L}_{\gamma,n} + \frac{(2b + X^2)^2}{8\gamma b}\sum_{i=1}^{K}\|\mathbf{u}_i\|^2\right]$$
$$+ \left(2\frac{\lambda}{b} - 1\right)\mathbb{E}\left[\sum_{i=1}^{K}\sum_{t=1}^{n} a_i A_{i,t}\right], \tag{9}$$

which concludes the proof. ∎

## A.2  Extension to $\kappa$ Queries per Round

We now allow the algorithm to query $\kappa$ labels instead of one. On each iteration $t$, the modified algorithm samples without repetitions $\kappa$ labels to be annotated, and perform the same update as of Eq. (2). Formally, on each round we have $\sum_i Z_{i,t} = \kappa$ for $Z_{i,t} \in \{0, 1\}$ where the first task-index to be queried is drawn according to Eq. (1). The second task is drawn from the same distribution, not allowing the first choice, and so on. Once $\kappa$ tasks are drawn, the algorithm receives $\kappa$ labels for the $\kappa$ corresponding inputs, and updates the $\kappa$ models according to Eq. (2).

**Corollary 4** *If SHAMPO algorithm gets feedback for $\kappa$ tasks on each round, instead of only a single task, the expected cumulative weighted mistakes is bounded as follows*

$$\mathbb{E}\left[\sum_{i=1}^{K}\sum_{t=1}^{n} M_{i,t}\right] \leq \frac{\delta}{\gamma\kappa}\left[\left(1+\frac{X^2}{2b}\right)\bar{L}_{\gamma,n}^{\kappa} + \kappa\frac{\left(2b+X^2\right)^2}{8\gamma^2 b}U^2\right] + \left(2\frac{\lambda}{b}-1\right)\mathbb{E}\left[\sum_{i=1}^{K}\sum_{t=1}^{n} a_i A_{i,t}\right] \ ,$$

*where $\bar{L}_{\gamma,n}^{\kappa}$ is the expected loss of $K$ models $\{\mathbf{u}_i\}$ over the $\kappa$ annotated instances per round $t$.*

**Proof:** We follow the proof of Thm. 1 until the end of the proof. We repeat the process $\kappa$ times, and get the equivalent inequality for sampling $\kappa$ tasks without repetitions, where $\delta_j$ is the per repetition quantity, and we have, $\delta = \max_j \delta_j$,

$$\left(\sum_{j=1}^{\kappa}\frac{1}{\delta_j}\right)\mathbb{E}\left[\sum_{i=1}^{K}\sum_{t=1}^{n}\left(M_{i,t} + \left(1-2\frac{\lambda}{b}\right)A_{i,t}a_i\right)\right] \leq \frac{1}{\gamma}\left[\left(1+\frac{X^2}{2b}\right)\bar{L}_{\gamma,n}^{\kappa} + \kappa\frac{\left(2b+X^2\right)^2}{8\gamma b}U^2\right] \ ,$$

(10)

where all expectations are now with respect to the sampling with repetitions, and specifically $\bar{L}_{\gamma,n}^{\kappa}$ is the expected loss of a set of linear models $\{\mathbf{u}_i\}$ where $\kappa$ tasks are sampled rather than a single one. For a choice of $\kappa = 1$ we get the bound of Thm. 1, as expected. ∎