[Reviews · NeurIPS 2014]

Submitted by Assigned_Reviewer_6

The paper presents a method for learning multiple tasks in parallel where at each round a sample is given per each task, but only a single task can have its sample annotated. The authors formulate their method using a trade-off between exploitation and exploration. Th1 provides an upper bound on the expected cumulative number of mistakes. The algorithm is compared to 2 different approaches for choosing the single sample/task to be annotated.

I like the paper. It is well written and provides good theoretical as well as experimental results.

I'm not sure the strict synchronic assumption on choice of annotation is really important. It seems more natural just to assume a total budget on annotation among all tasks in a learning system. It would be nice if the authors could add more real life motivation for this specific setting.

I also feel the experimental results could be strengthen by adding two more comparisons:
a. The case were annotation is cheap and all samples can be annotated. Obviously the results could be better but this will show the trade-off between being cheap on annotation or regarding the annotation as cheap.
b. An active learning approach applied to each task separately, while controlling the number of annotations among all tasks to be equal to the number of learning rounds.
This will provide a comparison to a method which "pays" the same on annotation but considers a weaker constraint among all tasks. Only the total number of annotation would have to be the same but each round several tasks could be annotated (or non).

Summary: I liked the paper, it presents an interesting new problem while providing nice theoretical and experimental guarantees for the proposed solution. I would like to see more motivation into the specific choice of parallel task learning together with experimental evidence to its justification (see suggestions above).

Submitted by Assigned_Reviewer_19

The paper introduced a novel multi-task framework where multiple online learners are sharing a single annotator. The proposed algorithm for the task is a perceptron like algorithm with an exploit and exploration strategy to decide the task to query for. The paper also includes an analysis of the proposed algorithm which bound the expected errors. The authors show that the algorithm can also be used to solve two different bandit problems. Empirical studies on standard machine learning dataset outperform weak baselines using naive query selection strategy under the same framework. The paper is well written. However, I found the proposed algorithm does not really leverage the relationship between different tasks to facilitate the learning, which in some senses are not as interesting as other multiple task learning algorithms. Moreover, the paper lacks an analysis on the sampling complexity of the proposed algorithm similar to the selective sampling literatures, which is very important for this kind of setup.

Detailed comments and suggestions:
1. The authors could include results for a full informative supervised learning baseline, it can help the reader learn the difficulty of the tasks.
2. What is the advantage of learning the tasks together under the proposed framework compared to learning the tasks separately with K selective sampling algorithms? It will make the paper more interesting to discuss this matters in the paper, as well as show empirical results to prove the points.

Summary: The paper introduced a novel multi-task framework and proposed a perceptron like algorithm with theoretical guarantee to solve the task. A major drawback of the proposed algorithm is that it does not leverage the relationship between the tasks to facilitate the joint learning.

Submitted by Assigned_Reviewer_42

This paper studies the problem of online multi-task learning in which a single annotator with limited bandwidth is shared among tasks. An online algorithm is proposed and its performance is analyzed.

Major Strengths:

Online multitask learning is an important topic, and the field of multi-armed bandits has a rich history. Seeking to leverage ideas and results from the latter to apply to the former domain will likely yield improved techniques, especially since for the multi-task learning problem some labels might not be observed. Furthermore, this area is highly relevant to NIPS.

Major Weaknesses:

An important aspect to multi-task learning is that there is some dependency between the tasks so that there is utility between learning the tasks jointly. In the work under review, no dependency is assumed. This appears to be a significant departure from the main multitask learning literature with no apparent justification. Without the ability to transfer knowledge between the tasks, it appears more as the multi-armed bandit analog in multi-task learning. For instance, why can only one feature be annotated at a time though all features are allowed to be annotated? In using Mechanical Turk, as suggested, for some applications the workers might not annotate labels for all features equally well. Thus, only allowing a subset of features to be annotated while the rest cannot seems more relevant of a setting. The article mentions it is a new multi-task framework, so it need not follow traditional assumptions, though more motivation and justification is needed for why the differences are important.

One counter-example to the previous statement is Romera-Paredes et al. “Exploiting Unrelated Tasks in Multi-Task Learning” in AISTATS 2012. There they use the knowledge that two tasks are unrelated to avoid overfitting by using the same features. However, the work under review does not exploit the lack of dependence.

This is less important, but to check the implication (lines 153-154) of Theorem 1, for the data analysis it would have been interesting to see how SHAMPO performs compared to the case when labels are available for all tasks.

Minor notes:

[25] “both allows”
[209] “outputs multicalss"
[347] remove comma
Summary: A new problem in multi-task learning is studied with detailed analysis.
Author Feedback
Author rebuttal: We thank the reviewers for the insightful comments.

R1+R2: Motivation and Justification. Indeed, our setting is different and our algorithm does not really leverage the (SIMILARITY) relationship between different tasks to facilitate the learning, yet it does implicitly leverage that some tasks are easier than others, and that some inputs are easier than other. Easy (and hard) are measured by the notion of margin. In this sense it is unique and original as we are not aware of any previous similar work. See lines 38-40 (first paragraph of intro). We explore this fact, and focus the labeling in the harder cases. We will make this intuition clearer in the final version.

R1:
Sampling complexity is irrelevant to our setting, as on each iteration our algorithm uses a single sample. It is not about if to query or not, but which query to perform.

R1+R2: We will try to include the full information supervised baseline. Note the results of Uniform are conceptually close to it, as in both cases we use random dataset, but in Uniform it is (1/K) the size of the full one. Thus, effectively, with with less data.

R1: In our model there is one query per round, yet when learning K selective sampling algorithms it is not guaranteed how many queries (or labels) will be used, as now these decisions are independent. We will try to add the results of K selective sampler in the final version.

R2: Thanks for the Romera-Paredes et al reference, we will add it to the final version.

R3: synchronic assumption on choice of annotation ...(vs) a total budget on annotation among all tasks in a learning system. It would be nice if the authors could add more real life motivation for this specific setting.
> Our assumption has a budget on the annotation "rate", as we work in online learning, where annotation and learning are performed on each step. We assume a budget per iteration. We believe that it is realistic in interactive systems. We will make it clearer in the final version.

R3: I also feel the experimental results could be strengthen by adding two more comparisons.
> We will try to add the two methods to the final version.